

# Comparison of methods of alert acknowledgement by critical care clinicians in the ICU setting

Andrew M. Harrison[1], Charat Thongprayoon[2], Christopher A. Aakre[3], Jack Y. Jeng[4], Mikhail A. Dziadzko[2], Ognjen Gajic[5], Brian W. Pickering[2] and Vitaly Herasevich[2]

[1] Medical Scientist Training Program, Mayo Clinic, Rochester, MN, United States of America
[2] Department of Anesthesiology, Mayo Clinic, Rochester, MN, United States of America
[3] Department of Internal Medicine, Mayo Clinic, Rochester, MN, United States of America
[4] Mayo Medical School, Mayo Clinic, Rochester, MN, United States of America
[5] Division of Pulmonology and Critical Care Medicine, Mayo Clinic, Rochester, MN, United States of America

Corresponding author
Vitaly Herasevich,
Herasevich.Vitaly@mayo.edu

## ABSTRACT

**Background.** Electronic Health Record (EHR)-based sepsis alert systems have failed to demonstrate improvements in clinically meaningful endpoints. However, the effect of implementation barriers on the success of new sepsis alert systems is rarely explored.
**Objective.** To test the hypothesis time to severe sepsis alert acknowledgement by critical care clinicians in the ICU setting would be reduced using an EHR-based alert acknowledgement system compared to a text paging-based system.
**Study Design.** In one arm of this simulation study, real alerts for patients in the medical ICU were delivered to critical care clinicians through the EHR. In the other arm, simulated alerts were delivered through text paging. The primary outcome was time to alert acknowledgement. The secondary outcomes were a structured, mixed quantitative/qualitative survey and informal group interview.
**Results.** The alert acknowledgement rate from the severe sepsis alert system was 3% ($N = 148$) and 51% ($N = 156$) from simulated severe sepsis alerts through traditional text paging. Time to alert acknowledgement from the severe sepsis alert system was median 274 min ($N = 5$) and median 2 min ($N = 80$) from text paging. The response rate from the EHR-based alert system was insufficient to compare primary measures. However, secondary measures revealed important barriers.
**Conclusion.** Alert fatigue, interruption, human error, and information overload are barriers to alert and simulation studies in the ICU setting.

# INTRODUCTION

Electronic health record (EHR)-based, automated sepsis alert systems have failed to demonstrate improvements in clinically meaningful endpoints, such as Intensive Care Unit (ICU)/hospital length of stay (LOS) and mortality (*Hooper et al., 2012*; *LaRosa et al., 2012*; *Nelson et al., 2011*; *Sawyer et al., 2011*). This includes studies of ICU-specific and non-ICU

specific alert systems. This also includes studies ranging in variation regarding degree of distinction between the detection of sepsis, severe sepsis, and/or septic shock (*Dellinger et al., 2013*). Clinically meaningful endpoints range from compliance with the international Surviving Sepsis Campaign (SSC) guidelines to hospital LOS, ICU LOS, and mortality. There are ICU-based and hospital wide means to trigger an alert for the early recognition of sepsis. Most EHRs now have a built in system to support this alert

Time to alert acknowledgement has been validated as one proxy for time to recognition of sepsis by critical care clinicians (*Dziadzko et al., 2016*). The failure of EHR-based, automated sepsis alert systems to be directly correlated with improvements in clinically meaningful endpoints is frequently attributed to limitations of detection algorithms and/or the need for clinical decision support (CDS) systems (*Semler et al., 2015*). Human factors, such as the impact of workflow changes or the influence of method of alert delivery, are known to be barriers to the implementation of new alert systems in the clinical setting (*Harrison, Herasevich & Gajic, 2015*).

As "alarm hazards" have been ranked as the top health technology hazard in the United States (*ECRI-Institute, 2013*), it is important to explore the effect of implementation of new alert systems on workflow changes and other human factors in the clinical setting. Despite outcome improvements in recent decades (*Kaukonen et al., 2014*), sepsis remains one of the most expensive in-hospital conditions (*Torio, 2013*). As one of the most technologically sophisticated hospital environments, the critical care setting serves as a model to explore the impact of implementation of new alert systems. Despite these technological advances, including widespread utilization of text messaging and smartphones by clinicians, text paging remains standard practice in the hospital setting (*HIMSS Analytics, 2016*). We hypothesized that time to severe sepsis alert acknowledgement by critical care clinicians in the ICU setting would be reduced using an EHR-based alert acknowledgement system compared to a text paging-based system.

## METHODS

### Study design and setting

This study was performed in February 2015 in the medical ICU at Mayo Clinic in Rochester, MN, USA (Fig. 1). This study was not performed in a laboratory setting or simulation environment. This medical ICU has been described previously as a benchmark for evaluation of institutional performance (*Afessa et al., 2005*). Severe sepsis alerts were delivered to critical care clinicians, including attending physicians, fellows, residents, and nurse practitioners/physician assistants (NPs/PAs) using traditional HIPAA-compliant text paging. This study was approved by the Mayo Clinic Institutional Review Board (IRB) for clinician-participant enrollment by oral consent.

### Study participants and medical ICU workflow

The medical ICU at Mayo Clinic consists of two physically adjacent 12-bed units, in close proximity to a nearby 9-bed respiratory care unit (RCU). For any given month, there are approximately 15 critical care attending physicians, six critical care fellows, four postgraduate year three internal medicine residents (PGY-3), six PGY-1 interns, and nine

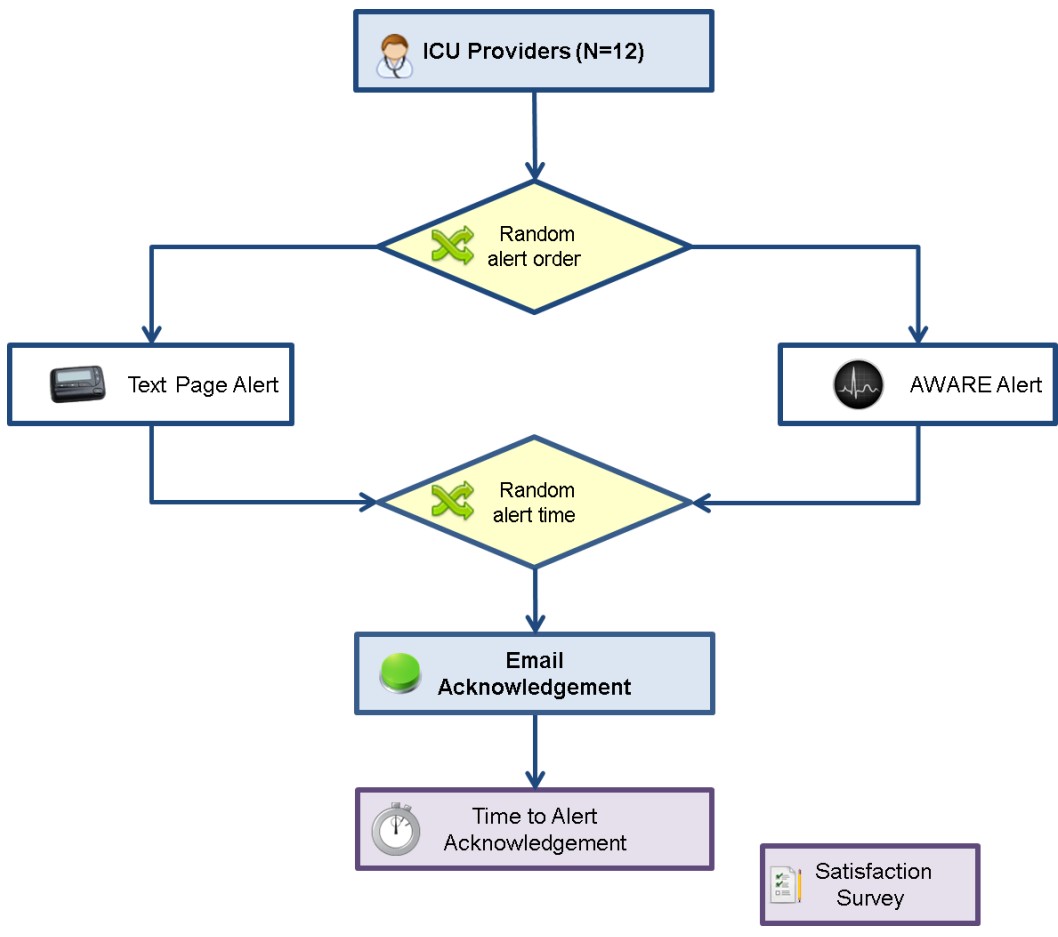

**Figure 1** Schematic illustration of study design.

dedicated medical ICU NPs/PAs. There are 2 shifts: 6 am–6 pm (AM) and 6 pm–6 am (PM). On any given day, the morning shift is further divided into two teams. Team 1 is assigned to the majority of the medical ICU patients. Team 2 is assigned the remaining patients, as well as the RCU, which is further staffed by an additional fellow and dedicated NP/PA from the same group of approximately 40 clinicians in the medical ICU that month.

## AWARE (Ambient Warning and Response Evaluation)
AWARE is the ICU-specific EHR system used in this study for patient viewer/monitoring. It was developed at Mayo Clinic and has been in routine clinical use in the medical ICU at Mayo Clinic since July 2012 (*Pickering et al., 2015*; *Pickering et al., 2010*). AWARE has been demonstrated to improve clinician task load, errors of cognition, and performance (*Ahmed et al., 2011*). AWARE is accessible from every computer workstation in this medical ICU, including bedside desktops, nursing stations, and clinician workrooms.

## Severe sepsis alert system
The severe sepsis detection algorithm was developed at Mayo Clinic and implemented into AWARE in December 2014 (*Harrison et al., 2015*). Within AWARE, the severe sepsis

Subject:    Sepsis Alert Research Study - SUNDAY

Greetings,

Thank you for agreeing to participate in our study (IRB 13-003325) to test mechanisms of alert delivery (text page vs EMR-Aware).

- You will receive 1-3 random text pages per shift in MB 6B/G for the next 4 weeks. Upon receipt, please directly email me with these Alert Numbers. Feel free to use this email as a reference point for responding, but please do not Reply All.
- You are also asked to email me the patient ID and icon color every time you identify a new* Yellow or Green sepsis alert icon in AWARE for *any* patient in MB 6B/G during your shifts, *even if this patient is not assigned to your care team*. *Please include icons already active at the start of your shift.
- One research restriction: we ask that you do not email via smartphone. Desktop, laptop/COW, and iPad are all acceptable.

I will not respond to your acknowledgement emails, but please feel free to contact me directly if there are any questions or concerns at any point throughout the duration of this study. At the conclusion of this study, we will coordinate a pizza party through the Division of Critical Care to celebrate.

**Figure 2** **Detailed daily email reminder to clinician participants with complete instructions.**

alert system displays a passive, yellow alert icon when severe sepsis is detected. This yellow alert icon can also be activated manually by clinicians for specific patients, when severe sepsis is suspected, but not detected by the automated alert system. This yellow alert icon is automatically updated to display a passive, green alert icon within AWARE after completion of the four elements of the 3-hour Surviving Sepsis Campaign (SSC) bundle (*Dellinger et al., 2013*). Once activated, the yellow alert icon will persist for at least 6 h, unless completion of the 6 h SSC bundle is detected (green alert icon) or manual deactivation by clinicians occurs. In the context of prolonged severe sepsis and/or septic shock, the yellow alert icon can persist indefinitely. The green alert icon automatically reverts back to "no sepsis detected" after 3 h, unless additional automatic (or manual) activation occurs.

## Study procedures

Clinicians agreed to participate in this study from February 02 through February 28 in 2015. The evening before each AM shift and the next PM shift, clinician participants for these upcoming shifts received a detailed email reminder with instructions (Fig. 2). The number of severe sepsis system alerts per shift through AWARE (yellow or green icon alerts) was entirely dependent on the number of septic patients in the medical ICU during any specific shift. Clinician participants randomly received no more than 3 simulated severe sepsis alerts per shift via text paging. In both cases, clinician participants were instructed to acknowledge all AWARE and traditional text paging severe sepsis alerts by email response. The difference between the time to severe sepsis alert activation in AWARE (or alert delivery via text page) and email response was defined as the time to alert acknowledgement.

## Survey design

To compare clinician satisfaction between the EHR-based alert acknowledgement system and text paging-based system, clinician participants completed a structured, mixed quantitative/qualitative survey, upon completion of the severe sepsis alert acknowledgement portion of this study (full survey facsimile in Results, Fig. 3). These questions were designed partially on existing clinician satisfaction surveys of alert methods for use in the hospital and critical care settings (*Embi, Jain & Harris, 2008*; *Wagner et al., 1998*).

Please provide at least one suggestion for improving alert/notification delivery:
Attending 01: Having a nurse pay closer attention to the patient and be less involved in online shopping/facebook checking (something I see way too often) and not to hesitate to notify the clinician when something is not right. Sometimes when I pass by, there is no one around and patient is hypotensive and looks very ill, and no one is aware of it because nurse is on a break and the other nurse doesn't know the patient and is otherwise occupied.
Attending 02: Text page for urgent issues should work. For non-urgent issues, creating more "in-baskets" or messages will create fatigue
Attending 03: Text message to phone or pager (phone preferred). Alternative would be a notification to phone through Synthesis or whatever the future app EMR might hold. This would be the most preferred for those of us that don't live glued to scanning the AWARE home screen.
Fellow 01: When notifying with pager add patient details too like room number, clinic number etc
Fellow 02: Most of residents do not use AWARE regularly. AWARE compliance need to be improved if messages are to be delivered by AWARE
Resident 02: I prefer pages. I'm still not used to using AWARE. If there is a truly urgent matter, I prefer a phone call to the portable phone.
NP/PA 01: I like paging as I always have it with me. I don't always have AWARE even with my iPAD that I have to log in to every 5-10 minutes.
NP/PA 02: No E-mial
NP/PA 03: Please use the pager for all urgent clinical alerts. Providers do not always have their Ipads or access to a computer, particularly during when spending time in patient rooms and in discussion with families.
NP/PA 04: Paging is best as I do not always have up e-mail if with patients, etc.. Text pages worked well. Just for thought: what if page did not go through… should there be a second option in place? **Duplicate Page ** notification
NP/PA 05: It is more difficult to keep up in the notifications through AWARE, especially if you don't always use the aware program throughout the day. I feel that the paging system would make me more prone to look at aware for a sepsis alert, but I feel that it would become a bit much and put to the side if I received a page for every sepsis alert everyday shift, all day. A suggestion would be to page with red alerts only or if no one has "claimed" a sepsis warning patient to remind us.

Please provide any additional comments you wish:
Attending 03: Native AWARE app would be great, with swiping between screens to move from organ system to organ system for those of us that are not in love with big clunky iPads … yes, even the ipad mini is too big for my taste.
NP/PA 01: Paging will get me alerted to soonest.
NP/PA 04: None, I hope my participation helped with the survey. I became more "aware" of the critically ill patients by participating – not only on my team but my colleagues' team as well. Became more mindful to offer assistance.
NP/PA 05: Due to my schedule during the trial period, I only worked a couple of shifts, which may have altered my input as I did not get accustom to it or have a lot of time working with the paging system. Thanks.

**Figure 3** Facsimile of the structured, mixed quantitative/qualitative survey provided to the clinician participants with all quantitative results overlaid: median (IQR).

## Statistical analysis

Severe sepsis alert system data was extracted directly from AWARE using METRIC Data Mart, a near-real time relational database of the complete EHR, which was developed at Mayo Clinic and has been described previously (*Herasevich et al., 2010*). Data was queried using JMP Pro (SAS Institute, Inc). Data collection and statistical analyses, such as the two-sided Student's $t$-test and the Chi-squared test, were also performed in JMP Pro. For all statistical analyses, a $p$-value of less than 0.05 was considered to be statistically significant. For all median values from the survey results, interquartile range (IQR) was reported.

**Table 1  Number of shifts per clinician participant and number of participants per shift.**

|  | Total AM Shifts | Total PM Shifts | Total Shifts | Shift, part 1 | Number of Providers | Shift, part 2 | Number of Providers |
|---|---|---|---|---|---|---|---|
| NP/PA #01 | 5 | – | 5 | 02/05 Thu PM | 1 | 02/12 Thu PM | 1 |
| NP/PA #02 | 4 | – | 4 | 02/06 Fri AM | 2 | 02/13 Fri AM | 4 |
| NP/PA #03 | 3 | 1 | 4 | 02/06 Fri PM | 2 | 02/13 Fri PM | – |
| NP/PA #04 | 3 | – | 3 | 02/07 Sat AM | 1 | 02/14 Sat AM | 6 |
| NP/PA #05 | 1 | – | 1 | 02/07 Sat PM | 2 | 02/14 Sat PM | – |
| Attending #01 | 7 | – | 7 | 02/02 Mon AM | 4 | 02/09 Mon AM | 2 |
| Attending #02 | 1 | – | 1 | 02/02 Mon PM | – | 02/09 Mon PM | 1 |
| Attending #03 | – | 1 | 1 | 02/03 Tue AM | 4 | 02/10 Tue AM | 5 |
| Fellow #01 | 7 | 2 | 9 | 02/03 Tue PM | 1 | 02/10 Tue PM | – |
| Fellow #02 | 6 | – | 6 | 02/04 Wed AM | 3 | 02/11 Wed AM | 5 |
| Resident #01 | 5 | 5 | 10 | 02/04 Wed PM | 1 | 02/11 Wed PM | – |
| Resident #02 | 7 | 2 | 9 | 02/05 Thu AM | 2 | 02/12 Thu AM | 4 |
|  |  |  |  | 02/08 Sun AM | 3 | 02/15 Sun AM | 4 |
|  |  |  |  | 02/08 Sun PM | 1 | 02/25 Sun PM | 1 |
|  | 49 | 11 | 60 |  | 27 |  | 33 |

# RESULTS

Prior to initiation of this study, a 1-day feasibility pilot was performed in January 2015 using seven medical ICU clinicians. Based on the result of this feasibility study (data not shown), it was determined that a sufficiently high clinician participant alert acknowledgement rate could be obtained from both severe sepsis system alerts through AWARE (yellow or green icon alerts) and simulated severe sepsis alerts through traditional text paging in the ICU setting. Based on the results of this feasibility pilot, participant instructions were optimized (Fig. 2).

Of the 40 clinicians staffing the medical ICU in February 2015, 13 (32%) were recruited to participate in this study. However, it was decided after two weeks (February 02 AM through February 15 PM) to prematurely terminate this study, due to sufficient statistical power for time to alert acknowledgement analysis, as well as feedback from clinician participants. As a result, it was necessary to exclude one NP/PA due to unavailability in the medical ICU during this shortened study period (RCU only). Ultimately, 12 clinicians participated: five NPs/PAs (out of nine), three attending physicians (out of 15), two fellows (out of six), two PGY-3s (out of four), and zero PGY-1s (out of six). The median number of potential AWARE alert acknowledgements per shift was two (IQR 1–4). The minimum and maximum numbers were zero and five. The number of patients who trigged at least 1 severe sepsis system alert through AWARE (yellow or green icon alert) was 28. Of the 28 shifts that occurred during this shortened study period, 23 shifts (82%) were covered by at least one participant (Table 1).

The alert acknowledgement rate from the severe sepsis alert system through AWARE was 3% ($N = 148$) and 51% ($N = 156$) from simulated severe sepsis alerts through text paging (Table 2). Time to alert acknowledgement from the severe sepsis alert system through

**Table 2 Comparison of alert response rate and median time to alert acknowledgement between the severe sepsis alert system through AWARE and simulated severe sepsis alerts through traditional text paging.**

|  | Text paging ($N = 156$) | AWARE ($N = 148$) | $p$-value |
|---|---|---|---|
| Alert response rate (N) | 51% (80) | 3% (5) | 0.001 |
| Median time to alert acknowledgement (IQR) | 2 mins (1–32) | 274 mins (130–517) | 0.053 |

AWARE was median 274 min ($N = 5$) and median 2 min ($N = 80$) from simulated severe sepsis alerts through text paging. The 5 alert acknowledgements from the severe sepsis alert system through AWARE came from only three clinician participants (NP/PA #01, NP/PA #04, and NP/PA #05), while all 12 participants acknowledged at least one simulated severe sepsis alert through text paging.

All participants completed a structured, mixed quantitative/qualitative survey. For the quantitative portion of the survey (Fig. 3), clinicians found alert by AWARE to be slightly less disruptive than alert by text paging. Clinicians found acknowledgement of AWARE and text paging alerts to be equally disruptive. When AWARE and text paging alerts were directly compared, a clear preference for text paging for both "urgent" and "non-urgent" alerts was present. When asked to "select one or more" (text paging, AWARE, email, phone call, text message, or other), the results for non-urgent alerts were mixed. However, when asked the same question for urgent alerts, the preference was once again clearly for text paging.

For the qualitative portion of the survey (Fig. 4), 11 out of 12 clinician participants provided "at least one suggestion for improving alert/notification delivery". Clinicians commented on inhomogeneous overall use of AWARE in the medical ICU, despite implementation several years prior (July 2012). Of the same 11 clinicians, four provided "any additional comments": the same three NPs/PAs who responded to at least one alert acknowledgement from the severe sepsis alert system through AWARE, as well as Attending #03. A clear theme concerning alert fatigue, interruption, human error, and information overload was present.

An informal group interview in the form of a noon pizza party was held to thank all clinician participants and gather additional feedback on the barriers to clinician participation and engagement in this implementation study. The four clinicians who attended were once again the same four clinicians who provided "any additional comments" on the survey. The statements regarding alert fatigue, interruption, human error, and information overload were reinforced, despite a strong interest from these clinicians to participate. Regarding inhomogeneous overall use of AWARE in the medical ICU, specific attention was drawn to a particular lack of interest from residents to use AWARE, as well as a lack of interest from both residents and fellows to participate in any research study during their required rotations through the medical ICU, including implementation of the severe sepsis alert system.

## DISCUSSION

We hypothesized that time to severe sepsis alert acknowledgement by critical care clinicians in the ICU setting would be reduced using an EHR-based alert acknowledgement system

Age (years): **41.0 (29.3 to 43.5)**
Years in practice: **6.5 (2.3 to 14.8)**
Years, months, weeks, or days working with the Mayo Clinic EMR: **2.5 years (1.1 to 13.3)**
Years, months, weeks, or days working in the medical ICU: **1.1 years (0.7 to 2.8)**
Years, months, weeks, or days working with AWARE: **1.4 years (0.6 to 2.5)**

Please rate 1 through 5:      1 (Never), 2 (Rarely), 3 (Sometimes), 4 (Frequently), or 5 (Always)
- Was notification by paging disruptive? **3 (2 to 3)**
- Was notification by AWARE disruptive? **2 (1 to 2)**
- Was paging acknowledgment difficult? **3 (1 to 4)**
- Was AWARE acknowledgment difficult? **3 (1 to 4)**

Please rate 1 through 5:      1 (Always paging), 2 (Mostly paging), 3 (No preference), 4 (Mostly
                              AWARE), 5 (Always AWARE)
- Which would be your preferred method of non-urgent alert/notification? **2 (1 to 5)**
- Which would be your preferred method of urgent alert/notification? **1 (1 to 2)**

For the questions below, if multiple options are preferred, please select more than one.
- The best method for non-urgent clinical alert/notification is (circle or **bold**)
  Paging **(6)**   AWARE **(5)**   Email **(3)**   Phone call **(0)**   Text message **(2)**   Other: **none (1)**
- The best method for urgent clinical alert/notification is (circle or **bold**)
  Paging **(11)**   AWARE **(1)**   Email **(0)**   Phone call **(2)**   Text message **(1)**   Other: **(0)**

Please provide at least one suggestion for improving alert/notification delivery:

**11 out of 12 participant provided a response.**

Please provide any additional comments you wish:

**4 out of the 11 participants above also provided an additional comment.**

**Figure 4**   **All qualitative responses to the structured, mixed quantitative/qualitative survey reproduced in their entirety, including typographical errors.**

compared to a text paging-based system. Based on the limited alert acknowledgement response rate using the severe sepsis alert system compared to traditional text paging, it was not possible to answer this hypothesis. However, feedback from the structured, mixed quantitative/qualitative survey, as well as the informal group interview, provided invaluable insight into the sources of this limited acknowledgement response rate. Implementation barriers included human factors, such as alert fatigue, interruption, human error, and information overload.

With the implementation of increasingly sophisticated EHR systems, interest in the development of novel automated detection and alert systems has increased (*Bourgault et al., 2014*). However, investigation into best methods of alert delivery (text paging, EHR systems, email, phone calls, and/or text messaging) for urgent and non-urgent alerts in the hospital setting is limited (*Gill, Kamath & Gill, 2012*). Investigation into the most appropriate clinician for alert delivery is also limited (*Zhang et al., 2003*). Monitoring and alert systems have been developed for patient use in the home setting (*Steinman et al., 2011*; *Tchalla et al., 2012*). However, there has been comparatively limited investigation into methods of alert delivery to clinicians in the hospital setting (*Loo et al., 2011*). Interestingly, many of these studies have been performed in the geriatric patient population, but not in

the ICU setting, where the average patient age is often 65 or older (*Seferian & Afessa, 2006*). Thus, there is a clear need for further systematic exploration of human factors barriers to the implementation of new alert systems in the ICU setting, such as the impact of workflow changes and the influence of method of alert delivery.

Implementation of automated detection and alert systems without consideration of these factors is known to have the potential to result in alert fatigue (*Singh et al., 2013*), interruption (*Hodgetts & Jones, 2007*), human error (*Bates et al., 1998*), and information overload (*Stokstad, 2001*). Recognition of the importance of alert fatigue in the hospital setting has increased significantly in recent years (*Herasevich et al., 2013*). However, implementation of automated alert systems generally must be performed in the context of information overload and complex task interruption (*Eppler & Mengis, 2004*). It is also known that information overload can alter alert perception in the medical setting (*Glassman et al., 2006*). This can cause clinicians to perceive alert systems negatively and deter future use (*Harrison et al., 2016*). Thus, the task of generating clinically meaningful alerts while concurrently minimizing information overload and task interruption is challenging.

Clinician-participant comments provided valuable insight regarding preferences for method of alert delivery. Although there was a clear preference for receiving urgent alerts through text paging, additional investigation is required to specifically explore the rationale for this preference. Understanding the rationale for this preference may reduce the barriers to answering the primary objective of this study, which was comparison of time to severe sepsis alert acknowledgement methods by critical care clinicians in the ICU setting. These secondary outcomes revealed important barriers to the inability to answer the primary outcome, which are applicable and generalizable to future studies.

Even the reluctance of clinicians to participate in this research study, as revealed during the post-study discussion, has the potential to confound interpretation of the results of this study. Whether this reluctance is the result of information overload, the Hawthorne effect, and/or some other factor(s), clinicians are currently struggling to find a balance between understanding of the importance of their participation in clinical research studies and "study fatigue."

This study has several limitations: (1) This was a single-center study at an academic medical center. Well-established biases and potential confounders are known to be present with this particular study design (*Straus et al., 2005*); (2) Unlike the severe sepsis system alerts through AWARE (yellow or green icon alerts), the severe sepsis alerts through text paging were simulated. Comparing non-simulated alerts to simulated alerts may introduce additional confounders into the interpretation of the results of this study; (3) Although not investigated in this study, the feasibility of severe sepsis alert delivery using an EHR-based, automated mobile app for smartphones has been validated (*Dziadzko et al., 2016*); (4) The significant range of clinical experience of clinician-participants introduces study bias. The potential application of this technology for the future of clinical practice and clinical research should not be ignored. Ultimately, a multi-center, non-simulated study in the ICU setting is required to address various aspects of these limitations.

## CONCLUSION

It could not be determined whether an automated alert for severe sepsis reduced time to alert acknowledgement by critical care clinicians in the ICU setting compared to text paging. This was due to an extremely limited alert acknowledgement response rate using the severe sepsis alert system compared to traditional text paging. Implementation barriers, including human factors—such as alert fatigue, interruption, human error, and information overload—were determined to be an important source of this finding.

### Funding

This work was supported by Center for Medicare and Medicaid Innovation (CMMI) Health Care Innovation Award "Patient Centered Cloud-based Electronic System: Ambient Warning and Response Evaluation (ProCCESs AWARE)" (1C1CMS330964-01-00). AMH is supported by a doctoral dissertation grant from the AHRQ (R36 HS022799). The funders had no role in study design, data collection and analysis, decision to publish, or preparation of the manuscript.

### Grant Disclosures

The following grant information was disclosed by the authors:
Center for Medicare and Medicaid Innovation (CMMI): 1C1CMS330964-01-00.
AHRQ: R36 HS022799.

### Competing Interests

AWARE is patent pending (US 2010/0198622, 12/697861, PCT/US2010/022750). Drs. Herasevich, Gajic, and Pickering and Mayo Clinic have a financial conflict of interest relating to licensed technology described in this paper. This research has been reviewed by the Mayo Clinic Conflict of Interest Review Board and is being conducted in compliance with Mayo Clinic Conflict of Interest Policies.

### Author Contributions

- Andrew M. Harrison performed the experiments, analyzed the data, wrote the paper, prepared figures and/or tables, reviewed drafts of the paper.
- Charat Thongprayoon, Christopher A. Aakre and Jack Y. Jeng performed the experiments, reviewed drafts of the paper.
- Mikhail A. Dziadzko analyzed the data, reviewed drafts of the paper.
- Ognjen Gajic, Brian W. Pickering and Vitaly Herasevich conceived and designed the experiments, reviewed drafts of the paper.

### Human Ethics

The following information was supplied relating to ethical approvals (i.e., approving body and any reference numbers):

This study was approved by the Mayo Clinic Institutional Review Board (IRB 13-003325).
**Patent Disclosures**

The following patent dependencies were disclosed by the authors:
US 2010/0198622, 12/697861, PCT/US2010/022750.

**Data Availability**

The raw data is included in the tables and figures of the manuscript.

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
