# Peer review of "Comparison of methods of alert acknowledgement by critical care clinicians in the ICU setting"

_PeerJ, doi:10.7717/peerj.3083_

## Round 0.1 · original submission · Major Revisions

These two reviews drew different conclusions from your paper. Reviewer 2's comments were relatively minor and easily addressed. Reviewer 1, however, expressed serious concerns about the experimental design.

Having read through the paper myself, I don't feel as strongly as reviewer 1, but I believe that this paper needs some substantial revision to be clear and comprehensible. For a paper that claims to be about barriers, there is very little discussion about what those barriers are. I suggest that this is paper is really an evaluation of the provision of alerts via the AWARE system as opposed to through text paging. However, it was very hard for me to understand which version was "automated", what the default/familiar delivery method was, and what exactly was being simulated. These key questions impact significantly on the readability of the paper.

Addressing these issues would significantly increase the readability of this paper. Please also address Reviewer 1's comments -both from the review itself and from comments in the text.

·

Basic reporting

This document adheres to being written in the right language with a reasonable degree of language flow and competency. Prior literature review is present in the body of the manuscript.

Experimental design

Experimental disease is fundamentally what this document is lacking. There is not a clear hypothesis and the ambigious one that exists is followed by nebulous methods with results and conclusions that often dont resemble anything close to the hypothesis. I found my self reading and re-reading sections to try to un derstand what was actually being considered and studied.

Validity of the findings

See above. I cannot well comment on the validity of findings when they are chaotic and don't resemble the elusive hypothesis.

Additional comments

The bulk of my comments are acutally in the body of the manuscript itself. Scientific method is challenging and, at times, cumbersome, and I commend the authors for attempting to complete this study. The topic of early sepsis alerts and the now nearly ubiquitous EHR is a good one and I encourage the authors to revise this study. Unfortunately, this manuscript does not contain a publishable scientific effort wherein a clear hypothesis is stated, two or more study groups are compared, and a difference (or lack thereof) is proven in the results and detailed in the discussion.

Reviewer 2 ·

Basic reporting

The authors have stated both that “the effect of implementation barriers on the success of new sepsis alert systems is rarely explored” and that “it is important to explore the effect of implementation of new alert systems on workflow changes and other human factors in the clinical setting” but neither seem to be clearly linked to the study’s hypothesis.

Experimental design

Include information regarding the location of the display for the AWARE monitoring system, e.g. patient bedside. As alert acknowledgment was via email response and email access may or may not be readily available in patient rooms, including this information would help to illustrate the study limitation noted regarding introduction of potential confounders.

Validity of the findings

No Comments.

Additional comments

The participants’ comments provided insight regarding their preferences for receiving alerts. The preference for receiving urgent alerts through text paging should receive more attention in the discussion and include the participants’ rationale for this preference.
Thank-you for the opportunity to review this manuscript.

---

## Round 0.2 · Minor Revisions

One reviewer has indicated that this revision is acceptable. As the reviewer who initially suggested "reject" has not reviewed this version, I have read the revision in detail. I concur that this version is much improved, but I will still insist upon consideration of a few key issues before continuing:

1. The hypothesis tested in this study is now more clearly stated, but I find that it is not consistent with the title. Generally, studies aimed at "barriers" are in-depth qualitative analyses examining workflows, contextual issues, and other factors influencing the success of a system. This paper presents a straightforward comparison between two approaches and some preliminary qualitative results, but it does not really discuss barriers. Please change the title to focus on the comparative nature of the study.

2. The paper implies, but does not state directly, that the text messaging system is current best practice. Please clarify - if it is not current practice, why was it chosen for this study?

3. In your discussion of the post-study discussion, you indicated a reluctance to participate in research. Might this be a confounding issue? Can you say more about this reluctance?

I have made a few other comments in the attached version of your paper. Please address these comments along with the three points mentioned above.

Reviewer 2 ·

Basic reporting

I was confused by lines 78-79. I am unfamiliar with the term “intuitional improvement perspective” and I could not find this term in the article that was cited. It would be helpful to define this term so that the reader can understand the information the authors are trying to impart with this sentence.

Experimental design

No comment.

Validity of the findings

No comment.

Additional comments

With the exception noted above, the revisions clarified several sections. Your findings will be of interest to the health care community.

---

## Round 0.3 · accepted · Accept

Thanks for your reviews. Your willingness to reconsider the title is appreciated - I think the paper is now much more readable.

One brief note about the reluctance of users to participate. It seems hard for me to understand how this reluctance could be a manifestation of the Hawthorne effect. You might consider either clarifying or removing this reference. I leave this to your discretion.

I'm looking forward to seeing the final published version of this paper.